



# Leaf carbon and nitrogen stoichiometric variation along environmental gradients

Huiying Xu[1], Han Wang[1], Iain Colin Prentice[1,2], Sandy P. Harrison[1,3]

[1]Department of Earth System Science, Ministry of Education Key Laboratory for Earth System Modeling, Institute for Global Change Studies, Tsinghua University, Beijing 100084, China;
[2]Georgina Mace Centre for the Living Planet, Department of Life Sciences, Imperial College London, Silwood Park Campus, Buckhurst Road, Ascot, SL5 7PY, UK;
[3]School of Archaeology, Geography and Environmental Sciences (SAGES), University of Reading, Reading, RG6 6AH, UK.

*Correspondence to*: Han Wang (wang_han@tsinghua.edu.cn)

**Abstract.** Leaf stoichiometric traits are central to ecosystem function and biogeochemical cycling, yet no accepted theory predicts their variation along environmental gradients. Using data in the China Plant Trait Database version 2, we aimed to characterize variation in leaf carbon and nitrogen per unit mass ($C_{mass}$, $N_{mass}$) and their ratio, and to test an eco-evolutionary optimality model for $N_{mass}$. Community-mean trait values were related to climate variables by multiple linear regression. Climatic optima and tolerances of major genera were estimated; Pagel's λ was used to quantify phylogenetic controls, and Bayesian phylogenetic linear mixed models to assess the contributions of climate, species identity and phylogeny. Optimality-based predictions of community-mean $N_{mass}$ were compared to observed values. All traits showed strong phylogenetic signals. Climate explained only 18% of C:N ratio variation among species but 45% among communities, highlighting the role of taxonomic replacement in mediating community-level responses. Geographic distributions of deciduous taxa separated primarily by moisture, evergreens by temperature. $C_{mass}$ increased with irradiance, but decreased with moisture and temperature. $N_{mass}$ declined with all three variables. C:N ratio variations were dominated by $N_{mass}$. The coefficients relating $N_{mass}$ to the ratio of maximum carboxylation capacity at 25 ˚C ($V_{cmax25}$) and leaf mass per area ($M_a$) were influenced by leaf area index. The optimality model captured 68% and 53% of variation between communities for $V_{cmax25}$ and $M_a$ respectively, and 30% for $N_{mass}$. We conclude that stoichiometric variations along climate gradients are achieved largely by environmental selection among species and clades with different characteristic trait values. Variations in leaf C:N ratio are mainly determined by $N_{mass}$, and optimality-based modelling shows useful predictive ability for community-mean $N_{mass}$. These findings should help to improve the representation of C:N coupling in ecosystem models.

## 1 Introduction

Nitrogen (N) has long been recognized as a key nutrient that influences photosynthesis, plant biomass and carbon (C) allocation, and therefore the terrestrial C cycle (Fernández-Martínez et al., 2014; Terrer et al., 2019). Many land surface models (LSMs) have recently incorporated representations of coupled C and N cycling, the intention being to increase the realism of model predictions of C cycling under climate change (Wiltshire et al., 2021). The leaf C:N ratio plays an essential role in this coupling; however it is often assigned a constant value per plant functional type (PFT), due to the lack of data and/or theory that would predict more realistic, continuous stoichiometric variation along environmental gradients (Meyerholt et al., 2020). One aspect of model uncertainty could be reduced if such variation were better understood and quantified (Boonman et al., 2020; Niu et al., 2023).

Many studies on leaf C and N along climate gradients have been carried out, but there is still no consensus on the major controls of such leaf traits at individual and community levels, hindering our understanding of trait-environment relationships (Anderegg, 2023). There is evidence for leaf stoichiometry being affected by many factors, including species identity, phylogeny, climate and soil properties (Elser et al., 2010; Ma et al., 2018; Tang et al., 2018; Yang et al., 2016). The





demonstrated roles of species identity and phylogeny indicate that leaf C and N contents ($C_{mass}$, $N_{mass}$) and their ratios are
phylogenetically conservative (Sardans and Penuelas, 2014; Vallicrosa et al., 2021; Zhang et al., 2012). Differences among
life forms and vegetation types have also been widely noted (Ma et al., 2018; Tang et al., 2018). On the other hand, few studies
have examined the differences between evergreen and deciduous leaves, which are expected to diverge as they represent
alternative life-history strategies, expressed in different responses of leaf mass per area ($M_a$) to climate (Kikuzawa et al., 2013;

Wang et al., 2023). The patterns of stoichiometric response to environment remain inconsistent across studies. A potential
contributory problem is their reliance on annual average climate variables, such as mean annual temperature – which does not
accurately reflect actual growing-season conditions, especially in regions with cold winters (Körner, 2021) – and mean annual
precipitation, which is generally not a good metric for plant-available moisture because it does not take account of the large
variations in potential evapotranspiration (driven by solar radiation and temperature) across the world.

50         Many land surface models (LSMs) treat leaf C:N ratios as fixed parameters for PFTs (Boonman et al., 2020; Zaehle
et al., 2014); some allow the C:N ratio to vary, within a prescribed range, based on C and N allocation to different tissues
(Ghimire et al., 2016; Meyerholt and Zaehle, 2015; Smith et al., 2014; Wang et al., 2010). But the fixed-PFT schemes fail to
capture the observed range of leaf stoichiometry within each PFT, while the dynamic schemes have not been extensively tested
against observations. Responses of N use efficiency and net primary production (NPP) to elevated $CO_2$ vary considerably

among models, and are not always realistic (Zaehle et al., 2014). Here eco-evolutionary optimality (EEO) principles may help
by providing a route towards testable, general trait predictions (Caldararu et al., 2020; Dong et al., 2022; Harrison et al., 2021;
Xu et al., 2021). Caldararu et al. (2020) applied an optimality-based approach (maximizing carbon export and growth) to
improve leaf $N_{mass}$ prediction, but this analysis did not consider the large and potentially confounding effect of $M_a$ variation
with environment (Wang *et al.*, 2023). We infer that there is still a need to investigate the eco-evoutionary basis of leaf C:N

ratio variations, and to reconsider how they are treated in LSMs (Sistla and Schimel, 2012).

In this study, we applied EEO principles to predict $N_{mass}$ from a trait-correlation perspective. We assumed that the
metabolic and structural components of leaf N are proportional to carboxylation capacity at a reference temperature of 25˚C
($V_{cmax25}$) and $M_a$, respectively. The coordination hypothesis provides predictions of $V_{cmax25}$: it is assumed that the light- and
Rubisco-limited assimilation rates under daytime conditions tend to equality, thus minimizing both maintenance respiration

and the metabolic component of leaf N (Chen et al., 1993). An EEO-based hypothesis for the leaf economics spectrum provides
predictions of $M_a$: it is assumed that the average net carbon gain by a leaf during its life cycle is maximized. The sum of the N
in metabolic and structural components then determines the optimal leaf N content ($N_{mass}$). We set out (1) to analyse the
contributions of climate, species identity and phylogeny to leaf $C_{mass}$, $N_{mass}$ and their ratio, (2) to characterize geographic
patterns in these traits along environmental gradients, (3) to test the extent to which variation in $N_{mass}$ among communities

could be captured by the EEO principles outlined above. Using a dataset comprising 1705 samples at 79 sites throughout China,
we quantified the phylogenetic signal in species' traits and fitted a Bayesian mixed-effects model to partition individual trait
variation into effects of climate, species identity and phylogeny. We examined trait relationships with bioclimate variables
(which improve on annual mean quantities by accounting for seasonality and latitude) and gridded data on soil C:N ratios by
multiple regression.

**2 Materials and methods**

**2.1 Trait and environmental data**

Our analyses are based on trait data in the China Plant Trait Database version 2 (CPTDv2, Wang et al. (2022)). The CPTDv2
contains morphometric, chemical and photosynthetic leaf trait data on 1529 species at 140 sites representing the different
biomes in China, as well as climate information for each site. A stratified sampling strategy was used to ensure that the

dominant species in each canopy layer were sampled (detailed in Wang et al. (2018)). A total of 25 trees, five shrubs, five



lianas or vines, and five understorey species (grasses and/or forbs) were sampled at each site. When the number of trees was less than 25 at a site, all the tree species were sampled and additional samples from the other life forms were supplemented up to a maximum of 40 species. Thus, the species sampled at each site can be considered as a representative sample of the plant community and average trait values at each site. We extracted leaf carbon concentration ($C_{mass}$, %), nitrogen concentration

($N_{mass}$, ‰), leaf mass per area ($M_a$, g biomass m$^{-2}$), stable carbon isotope ratios (δ$^{13}$C, ‰) and $V_{cmax25}$ (μmol C m$^{-2}$ s$^{-1}$) data from the CPTDv2. Although $M_a$ data are available from 124 sites, $C_{mass}$, $N_{mass}$ and δ$^{13}$C data are only available at 79 ($C_{mass}$, $N_{mass}$) and 74 sites, respectively. However, these sites are well distributed across the temperature and aridity gradients (Supplementary data Fig. S1). Although $V_{cmax25}$ data are available only at 32 sites from southwestern, northeastern China and along an elevational transect in Gongga Mountains, there are 960 measurements from these sites.

Xu et al. (2021) and Wang et al. (2018) provided full details of the species sampled and trait measurements made at each site. $M_a$ was estimated from measurements of leaf area and dry weight following standard protocols (Cornelissen et al., 2003). Leaf area was taken as the projected area of a randomly selected leaf, or leaflet for compound leaves, using a LiDE 220 Scanner (Canon Inc., Huntington, NY, USA). The dry weight was measured after oven-drying at 75 ˚C for 72 h to constant weight. The average of three measurements made on leaves from different individuals was taken as the $M_a$ value of one species

at each site. We used a portable infrared gas analyser system (LI-6400; Li-Cor Inc., Lincoln, NE, USA) to make leaf gas-exchange measurements in the field. Terminal branches from the outer canopy were collected and re-cut under water immediately prior to measurement. The relative humidity and chamber block temperature were set close to that of the ambient environment at the time of measurement with a constant airflow rate (500 μmol s$^{-1}$). $V_{cmax}$ was calculated from the light-saturated rate of CO$_2$ fixation at ambient CO$_2$ using the one-point method (De Kauwe et al., 2016) and adjusted to a standard

temperature of 25˚C with the Arrhenius equation (Bernacchi et al., 2001). Due to the time-consuming measurement of leaf-gas exchange, photosynthetic traits of one sample were measured for each species. Leaf C content, N content and δ$^{13}$C were measured using pooled samples of leaves from multiple (at least three) individuals with an Isotope Ratio Mass Spectrometer (Thermo Fisher Scientific Inc., Carlsbad, CA, USA). Carbon isotope ratios were used to calculate isotopic discrimination (Δ), and then to estimate the ratio of leaf-internal to ambient CO$_2$ partial pressure (χ) using the method of Cornwell et al. (2018)

with a standard formula using the recommended values of *a'* and *b'* of 4.4 ‰ and 27 ‰, respectively (Cernusak et al., 2013; Farquhar et al., 1989):

$$\chi = \frac{\Delta - a'}{b' - a'} \qquad\qquad (1)$$

The bioclimate variables available for each site include an annual plant-available moisture index ($\alpha_p$, an estimate of the ratio of annual actual evapotranspiration to potential evapotranspiration), mean temperature of the coldest month (MTCO, ˚C),

mean temperature during the thermal growing season, defined as the period with temperatures above 0 ˚C (mGDD0, ˚C), and leaf area index- (LAI-) weighted photosynthetic photon flux density ($I_{abs}$, mol m$^{-2}$ s$^{-1}$) during the thermal growing season. The climate variables were interpolated to each site from 1814 weather stations in China using ANUSPLIN (Hutchinson and Xu, 2004). The LAI during the sampled month and year for each site from both datasets was extracted from the MODIS LAI product (MCD15A3H: https://modis.gsfc.nasa.gov/) to provide a measure of canopy cover. We used the C:N ratio in topsoil

(0-30 cm) extracted from the gridded soil data set of Shangguan et al. (2013) as an inverse index of soil fertility.

## 2.2 Trait prediction

The maximum capacity of carboxylation ($V_{cmax}$) was predicted using an EEO model based on the coordination hypothesis (Eq. 2), which states that plants coordinate Rubisco-limited and light-limited photosynthesis rates to be equal under daytime conditions so that the available light is used without incurring futile maintenance costs (Prentice et al., 2014; Wang et al.,

120    2017):





$$V_{cmax} \approx \frac{\varphi_0 I_{abs}(c_a\chi + K)}{c_a\chi + 2\Gamma^*}$$  (2)

$$\varphi_0 = \frac{0.352 + 0.021T - 0.00034\,T^2}{8}$$  (3)

where $\varphi_0$ is the intrinsic quantum efficiency of photosynthesis (µmol C µmol$^{-1}$ photon), which can be estimated for C$_3$ plants using Eq. (3) (Bernacchi et al., 2003), $c_a$ is the ambient partial pressure of CO$_2$ (Pa), $\chi$ is the ratio of leaf-internal to ambient

CO$_2$ partial pressure (Pa Pa$^{-1}$), $K$ is the effective Michaelis-Menten coefficient of Rubisco (Pa), $\Gamma^*$ is the photorespiratory compensation point (Pa), and $T$ is temperature (˚C). We used mGDD0 as the temperature input.

Ma was predicted using an eco-evolutionary optimality model that predicts the relationship between $M_a$ and leaf longevity, based on the assumption that leaves maximize net carbon gain during their life cycle (Wang et al., 2023). The predicted environmental effects on $M_a$ differ between evergreen and deciduous species:

$$\ln(M_{a,de}) = \ln(f) + \ln(I_{abs}) - 0.052\,T - 0.27\ln(\alpha_p) + 2.65$$  (4)

$$\ln(M_{a,ev}) = 0.25\ln(f) + 0.5\ln(I_{abs}) - 0.013\,T - 0.51\ln(\alpha_p) + 3.53$$  (5)

where $M_{a,de}$ and $M_{a,ev}$ are the predicted $M_a$ for deciduous and evergreen species respectively, and $f$ is the ratio of thermal growing season length (days) to the number of days in the year.

Dong et al. (2017) proposed a model for $N_{area}$ as the sum of components proportional to $M_a$ and $V_{cmax25}$ respectively.

A simple manipulation of this model gives:

$$N_{mass} = a + \frac{b\,V_{cmax25}}{M_a}$$  (6)

where $a$ (unitless) and $b$ (g biomass g N s µmol C$^{-1}$) are empirical coefficients fitted across all species. To test whether nitrogen allocation varied within the canopy (Charles-Edwards et al., 1987), the random effect of binned LAI on the intercept ($a$) and slope ($b$) was tested using a mixed-effects model. The fitted values of $a$ and $b$ can be found in Supplementary data Table S2.

The C:N ratio was estimated as the ratio of predicted $C_{mass}$ from Eq. (7) and $N_{mass}$ from Eq. (6) (Fig. 6). We also calculated C:N ratios using observed $C_{mass}$ and predicted $N_{mass}$ (Fig. 6a) to check whether $C_{mass}$ values influence the prediction of C:N ratios. Due to the lack of an existing theoretical basis to predict $C_{mass}$, we fitted the following linear regression using all observed trait data and three climate variables:

$$\ln(C_{mass}) = 3.06\ln(I_{abs}) - 0.18\,T - 0.48\ln(\alpha_p)$$  (7)

**2.3 Data analysis**

Statistical analyses were carried out in R4.1.1 (R Core Team 2021). The relative importance of $C_{mass}$ and $N_{mass}$ in controlling the leaf C:N ratio was evaluated using the *relaimpo* package (Groemping, 2006). Within- and between-site variability in traits was measured by the standard deviation (SD). Bioclimatic effects on leaf stoichiometry at community level (i.e., with unweighted community-mean values as the data points) were examined using standard multiple (fixed-effects) linear

regression (*lm*) and partial effects of each climate variable were visualized using *visreg* (Breheny and Burchett, 2017). Phylogenetic analyses were carried out on all species (including 561 genera in 175 families and 57 orders). Phylogenetic trees were constructed and coloured with species-averaged trait values using the *S.PhyloMaker* and *ggtree* packages (Qian and Jin, 2016; Yu et al., 2017). Pagel's λ was calculated for each trait using the *phytools* package (Münkemüller et al., 2012; Revell, 2012). A Bayesian phylogenetic linear mixed model was applied at species level (i.e. each occurrence of each species was

treated as a data point), with species identity and phylogeny as random effects, using the *MCMCglmm* package (Hadfield,





2010). The model was repeated using three different phylogenetic hypotheses to account for the uncertainty of phylogenetic trees generated in these scenarios. Marginal (climate effects alone as fixed effects, without random effects) and conditional $r^2$ (with species and phylogeny as random effects) were compared (Nakagawa et al., 2017; Nakagawa and Schielzeth, 2013). The phylogenetic comparative method, which implicitly attributes overlapping effects of phylogeny and climate entirely to

phylogeny (Westoby et al., 1995), was used to estimate trait variations explained by climate alone using the *ape* package (Paradis et al., 2004).

The temperature and moisture optima and tolerances of frequently-occurring genera were calculated as follows (Meng et al., 2015). Each bioclimatic variable was binned and the mean abundance was calculated for the sites within each bin. The frequency distributions of abundance for each species and bioclimatic variable were obtained by selecting widths of the bins.

The optimum was estimated as the average of the bioclimatic variable in the bins where a species was present, weighted by its mean abundance in the bins. Similarly, the tolerance was estimated as the abundance-weighted standard deviation of the bioclimatic variable. Frequently occurring species were defined as those that occurred more than 25 times for deciduous and more than ten times for evergreen species, respectively.

### 3 Results

### 3.1 Roles of phylogeny and species in stoichiometric variation

In general, related species tended to have similar stoichiometric traits; Pagel's λ was significant for all traits (Fig. 1). The mixed model produced higher conditional $r^2$ values when species and phylogeny were included as random effects, with species contributing 21–35% and phylogeny contributing 16–18% on average (Table 1). No significant relationships were found between leaf stoichiometric traits and soil C:N ratio (Supplementary data Fig. S2).

These findings are consistent with species turnover (taxonomic replacement) being a principal mechanism accounting for the observed trends in stoichiometry along environmental gradients. Figure 2 illustrates the turnover of major woody genera along the climatic gradients. The deciduous genera covered a wide range in moisture ($\alpha_p$ from 0.2 to 1) and light ($I_{abs}$ from 7 to 16 mol m$^{-2}$ s$^{-1}$) but showed limited temperature tolerance ranges (Fig. 2a). Conversely, evergreen genera occupied a wide range on the temperature axis (from 6 to 21 °C) but occurred only in wetter areas with $\alpha_p$ > 0.6 (Fig. 2b). Growing-season

(mGDD0) and coldest-month (MTCO) temperatures were positively correlated (not shown). The distribution of deciduous genera along the MTCO axis was similar to their distribution along the mGDD0 axis (Fig. 2c), whereas evergreen genera were more separated on the mGDD0 gradient than by MTCO – with the exception of *Pinus*, which showed a wide cold-tolerance range from around –30 to 0 °C (Fig. 2d).

### 3.2 Leaf stoichiometric trait responses to climate

At community level, climatic variables explained 10%, 13% and 45% of variation in $C_{mass}$, $N_{mass}$ and C:N ratio respectively. At species level, climatic variables explained 8%, 3% and 18% according to the mixed model. Smaller amounts of variation (2%, 2% and 10%) were captured by climate according to the phylogenetic comparative method (Table 1).

Stoichiometric trait responses to climate were generally similar in deciduous and evergreen species (Fig. 3). $C_{mass}$ was significantly positively related to light and negatively related to moisture and growing-season temperature in both

deciduous and evergreen species (Fig. 3a-c). $N_{mass}$ significantly decreased with increasing light and moisture in both deciduous and evergreen species. $N_{mass}$ also decreased with temperature in deciduous species, but showed no significant relationship with temperature in evergreen species (Fig. 3d-f). The response of leaf C:N ratio to climate was a combination of the $C_{mass}$ and $N_{mass}$ responses, but was dominated by climate effects on $N_{mass}$. Leaf C:N ratio was positively related to light and moisture in both deciduous and evergreen species. It was also positively related to temperature for deciduous species, but marginally negatively

related to temperature for evergreen species (Fig. 3g-i).



### 3.3 Eco-evolutionary optimality models for leaf traits

$C_{mass}$ was relatively constant at different values of leaf C:N ratio (grey lines, Supplementary data Fig. S3), while $N_{mass}$ showed much greater variability. This pattern held for both deciduous and evergreen species. The analysis of relative importance showed that $N_{mass}$ explains on average 90% of variation in the leaf C:N ratio.

Leaf $N_{mass}$ was positively related to its theoretical predictor ($V_{cmax25}/M_a$) (Fig. 4). We found a significant LAI effect on the slope ($b$) and intercept ($a$) of this relationship, the slope increasing and intercept decreasing towards greater LAI. The $r^2$ of Eq. (6) was improved from 0.14 to 0.21 at species level after including the LAI effect. The optimality models captured 68% and 53% of the community-level variation in $V_{cmax25}$ and $M_a$, respectively (Fig. 5a,b). $V_{cmax25}$ was somewhat underestimated at most sites, with the largest bias when observed $V_{cmax25}$ was at alpine sites above 4000 m. $M_a$ was distributed
evenly near the 1:1 line, with the largest bias occurring at a semi-arid site with very high observed $M_a$. The optimality model, with LAI effect included, explained 30% of $N_{mass}$ variation using predicted values of $V_{cmax25}$ and $M_a$ (Fig. 5c). The predicted leaf C:N ratios fell within observed range in each PFT and outperformed fixed values prescribed in LSMs for most PFTs (Fig. 6). The prediction of leaf C:N ratio using constant $C_{mass}$ (45.6%) was similar to that using observed $C_{mass}$.

### 4 Discussion

We have demonstrated that across-site variations in leaf stoichiometric traits along climate gradients are driven mainly by species turnover; and that an optimality-based model can predict 30% of $N_{mass}$ variation, highlighting the potential of applying EEO principles to leaf stoichiometry. Predicted leaf C:N ratios are within the range of observations. These findings provide a potemtial avenue for improving the representation of leaf stoichiometry in LSMs.

### 4.1 Climate effects mediated by compositional shifts

For $C_{mass}$ and $N_{mass}$ separately, the contribution of climate variables in the multiple regression was modest (10–13%) while the mixed model attributed larger fractions of variation to species identity and phylogeny than to climate (Table 1). For the C:N ratio, climate explained 45% of variation in the multiple regression, while the mixed model attributed similar fractions (19–20%) of variation to species identity, phylogeny and climate (Table 1), indicating limited intraspecific trait variations. The phylogenetic comparative method (Table 1) attributed only 2% of variation in $C_{mass}$ and $N_{mass}$, and 10% of variation in C:N
ratio, to climate alone. These results are consistent both with strong phylogenetic control of leaf stoichiometry, and with strong patterns of variation (especially for C:N ratio) in community-mean values determined to a substantial degree by environmental selection among species and clades characterized by different trait values (Liu et al., 2022).

Within-site variations – unconstrained by macroclimate – were usually larger than between-site variations (Supplementary data Fig. S4). This large within-site variability might explain why no significant effect of climate on leaf
stoichiometry was detected in some previous regional studies (Yang et al., 2016; Zhang et al., 2017; Zhao et al., 2018). Zhang et al. (2019) showed a weak phylogenetic signal for the leaf C:N ratio evaluated by Blomberg's $K$. We used Pagel's $\lambda$ due to its better performance and reliability with large number of species (Münkemüller et al., 2012). The significant phylogenetic signals for leaf stoichiometric traits confirmed that species with similar evolutionary history tend to have similar leaf stoichiometry, indicating that leaf stoichiometric traits of extant species at a site may not remain adaptive under a changing
environment (He et al., 2010; Li et al., 2021; Yang et al., 2016). It has been suggested that a high plasticity of the leaf C:N ratio would be associated with a high mortality risk, supporting the idea that tight regulation of leaf stoichiometry within species helps to ensure plant survival (Luong et al., 2021). Within-site diversity may help communities to maintain their function in the face of climate variability and extremes.

Plant species may occupy different "biogeochemical niches", to ensure the full use of available resources and avoid
interspecific competition (Sardans and Penuelas, 2014; Sardans et al., 2021). At community level, climate variables captured



more of the observed leaf stoichiometric variations, due to the averaging of data from co-occurring species (Vallicrosa et al., 2021). Given the relative lack of plasticity in leaf stoichiometry, systematic variation in community-mean leaf stoichiometric traits along climate gradients can nonetheless be achieved through progressive species replacement (Liu et al., 2019; Yang et al., 2016).

The distributions of common deciduous genera were shown to be more sensitive to moisture, while the distributions of evergreen genera were mainly driven by temperature. This distinction may be related to the different adaptation strategies represented by differences in leaf longevity (LL). Kikuzawa et al. (2013) indicated that temperature is the best predictor of LL for evergreen species, while consideration of an additional moisture factor was expected to improve the explanatory power of climate for LL in deciduous species. According to Kikuzawa's optimality model, LL of evergreen species is higher at low

temperatures, in order to compensate for low total carbon gain during the short growing season. For deciduous species, however, LL should not exceed the length of the growing season – which can be affected by moisture as well as temperature in semi-arid and arid areas. Thus, our study suggests that climate shapes leaf stoichiometric variation via environmental selection among taxa, and emphasizes the neglected role of phenology in biogeochemical cycles (He et al., 2006; Vallicrosa et al., 2021; Xiong et al., 2021).

Although some studies have shown an important role of soil fertility in determining plant stoichiometry, published studies have shown inconsistent results (Fang et al., 2019; Fyllas et al., 2009; He et al., 2010; Ordoñez et al., 2009; Xiong et al., 2021). Soil fertility as indexed by the soil C:N ratio had no significant effect on leaf stoichiometry in our analysis, indicating a decoupling of soil and leaf stoichiometry (Delgado-Baquerizo et al., 2017; Elser et al., 2010). Plant-soil interactions may affect whole-plant stoichiometry nonetheless, through effects on C allocation to different tissues. Allocation of N to leaves

shows stronger homeostasis than other tissues, possibly as a consequence of the need to maintain the crucial functions of photosynthesis and leaf respiration; the stoichiometry of other tissues may adjust to soil conditions in order to support leaf-level function (Chen et al., 2013; Delgado-Baquerizo et al., 2017; Zhang et al., 2017).

### 4.2 Trait responses reflect plant strategies

Leaf stoichiometry integrates traits that reflect different plant functions, resulting in a potentially complex response to climate.

In contrast with many previous studies, we have considered (and found significant effects of) functionally significant bioclimatic variables, including light, on leaf-level stoichiometry. Our analyses indicate general relationships that are quite similar between evergreen and deciduous plants.

Higher leaf $C_{mass}$ was observed in cold and dry areas with high radiation (Fig. 3). Chen et al. (2021) found that leaf $C_{mass}$ is positively related to vein density, which relates to the efficiency of water transport. At high light plants tend to have a

higher photosynthetic rate, requiring more water for transpiration to maintain open stomata – which could be achieved by high carbon investment in venation (Sack and Scoffoni, 2013). In dry areas, high vein density is a common adaptation to drought, allowing plants to respond quickly to available water for carbon fixation, and to keep leaves cool in the face of high air temperature (Scoffoni et al., 2011; Yao et al., 2021). Meanwhile, plants may accumulate NSCs to adjust osmotic potential and avoid leaf desiccation (Bartlett et al., 2014). The leaf $C_{mass}$ response to temperature as observed here however is opposite to

some previous reports (Ma et al., 2018; Xing et al., 2021). Global analysis showed an overall positive response to temperature with range from –10 to 30 ˚C, whereas leaf $C_{mass}$ decreased when mean annual temperature was lower than 20 ˚C (Ma et al., 2018). $M_a$, which is positively related to leaf $C_{mass}$ (Xing et al., 2021), is generally negatively correlated with temperature (Wright et al., 2004). Higher starch concentration is observed at low temperatures, due to conditions that allow photosynthesis but not growth (Hoch and Körner, 2012). This suggests that leaf $C_{mass}$ response to temperature may not be monotonic, owing

to different functions dominating at the extremes.

The components of leaf N variation adapt to the climate in different ways (Dong et al., 2017; Peng et al., 2020; Xu et al., 2021). $N_{mass}$ is also constrained by the trade-offs inherent in the leaf economics spectrum. Leaves with high $M_a$ (and LL)





have low photosynthetic rates per unit mass, and low nutrient contents by mass (Wright et al., 2004). Thus, climate drives $N_{mass}$ variation both directly and indirectly. Moisture has a negative effect on $N_{mass}$ (Yang et al., 2016; Zhang et al., 2019; Zhao et al., 2018). It has been reported that N-containing compounds (such as amides) accumulate in plants in order to adjust osmotic pressure under drought (Raggi, 1994). In addition, N-rich leaf defence compounds increase towards more arid climates, at the expense of C-based defences such as spines and thorns (de Oliveira et al., 2020; Ghimire et al., 2017; Meloni et al., 2012). In the existing optimality model $N_{area}$ should be positively related to radiation, since light has a positive effect on $M_a$ and $V_{cmax25}$ (Smith et al., 2019; Wang et al., 2023). However, when N is expressed on a mass basis ($N_{mass}$), light has a negative effect, suggesting a lower sensitivity of $V_{cmax25}$ to light than $M_a$. The negative effect of temperature on $N_{mass}$ has been observed across vegetation types (Han et al., 2005; He et al., 2008; Tang et al., 2018; Weih and Karlsson, 2001). This is consistent with the hypothesis that more nutrients are required to compensate for low enzyme activity at low temperatures (Reich and Oleksyn, 2004). Although the leaf C:N ratio response to climate is a combination of the responses of both C and N, it is dominated by the variation of N. Thus, understanding of $N_{mass}$ variation should help elucidate variation in C:N ratios (Reich, 2005). Positive effects of temperature and moisture on leaf C:N ratios have also been observed in previous studies, implying higher N use efficiency in hot and wet areas (Fang et al., 2019; Zhang et al., 2019).

### 4.3 Leaf nitrogen content predicted by optimality models

C:N ratios couple C and N cycling, thus influencing the estimation of carbon assimilation and plant growth in LSMs (Wang et al., 2010; Zaehle et al., 2014). Fixed leaf C:N ratios assigned to PFTs, as for example in CLM4 and ED2.1, may result in inaccurate representations of this coupling (Bonan and Doney, 2018; Lawrence et al., 2011; Medvigy et al., 2009). Although model outputs such as ecosystem responses to elevated $CO_2$ are more consistent with observations in models where flexible C:N ratios are allowed (Lawrence et al., 2019; Meyerholt and Zaehle, 2015), large differences between models persist (Du et al., 2018).

Meyerholt and Zaehle (2015) highlighted the potential of optimality theory to improve the representation of N cycling in LSMs. Caldararu et al. (2020) showed that models that implement dynamic leaf stoichiometry schemes based on EEO principles can perform better than those with fixed-PFT schemes. Here we have shown that leaf $N_{mass}$ covariation with $V_{cmax25}$ and $M_a$ as predicted by EEO principles can provide further insights. Since $N_{mass}$ is the key to determining the leaf C:N ratio, given the relative constancy of $C_{mass}$ (Reich, 2005), we focused on the predictability of $N_{mass}$. The variation of leaf N per unit area ($N_{area}$) can be represented as the sum of two components, proportional to leaf mass per area ($M_a$) and the maximum capacity of carboxylation at 25 °C ($V_{cmax25}$) respectively (Dong et al., 2017); and now both $M_a$ and $V_{cmax25}$ can be predicted from EEO principles (Smith et al., 2019; Wang et al., 2023; Xu et al., 2021). Community-level variations in $M_a$, $V_{cmax25}$ and $N_{area}$ can indeed be largely captured ($r^2 = 0.53$, $0.68$, and $0.62$ respectively) using climate variables as predictors. We also showed a tendency for the relationship between $N_{mass}$ and the ratio $V_{cmax25}/M_a$ to become steeper with increasing LAI. This finding is consistent with N redistribution within the canopy, as an acclimation to light conditions that maximizes total carbon gain (Hirose and Werger, 1987; Niinemets et al., 2015). The strong vertical light gradient in high-LAI canopies implies a large advantage for optimized N distribution, in contrast with more open canopies (Field, 1983). The same model framework as $N_{area}$, with this additional LAI effect included, showed good predictive skill for $N_{mass}$ (and better than that of Boonman et al. (2020), obtained using an ensemble modelling approach) based on climate. Predicted C:N ratios, whether using observed or constant $C_{mass}$, lie within the range of observed data, supporting the dominant role of $N_{mass}$ in driving leaf C:N ratios (Fig. 6). The target (PFT-specific) values used in LSMs are based on datasets nearly 20 years old and fail to represent continuous trait variations that can now be inferred from much larger data sets. Our EEO-based approach this suggests a way forward to improve the dynamic representation of leaf stoichiometry in LSMs.

### 5 Conclusion


This study shows that leaf C:N ratio is mainly driven by mass-based leaf nitrogen content which can be estimated via the sum
of metabolic and structural components of leaf nitrogen using eco-evolutionary optimality-based models. This provides another
perspective to improve dynamic representation of stoichiometry in Earth System models. The variations in leaf stoichiometric
traits at individual level are mainly controlled by species identity and phylogeny, thus, the shift of leaf stoichiometry variations
at community level along climate gradient is achieved via species turnover. This allows the prediction of community-mean
values of leaf stoichiometric traits using EEO-based models. We show that the coefficient representing nitrogen allocation to
metabolic and structural components is related to leaf area index, which highlights the importance of nitrogen allocation in its
prediction. The unexplained variation in leaf nitrogen content may attribute to other unclear physiological processes which
requires further effort to improve the prediction of leaf C:N ratio.

**Data availability**

All        traits        and        climate        data        are        available        from        figshare
(https://figshare.com/articles/dataset/The_China_Plant_Trait_Database_Version_2_0/19448219).

**Authors' contributions**

H.X. carried out the analyses and prepared the manuscript with contributions from all co-authors. H.W. conceived the study
design. H.W., S.P.H. and I.C.P. contributed to the analyses and interpretation of the results.

**Competing interests**

The authors declare that they have no conflict of interest.

**Funding**

This work was supported by the National Natural Science Foundation of China (grant numbers 32022052, 31971495,
91837312, 2018YFA0605400). Participation of I.C.P. and S.P.H has been supported by the High-End Foreign Expert
programme of the China State Administration of Foreign Expert Affairs at Tsinghua University (grant numbers G20190001075,
G20200001064, G2021102001). I.C.P. acknowledges support from the European Research Council (787203 REALM) under
the European Union's Horizon 2020 research programme. S.P.H. is supported by the European Research Council (694481
GC2.0) under the same programme. This work is a contribution to the LEMONTREE (Land Ecosystem Models based On New
Theory, obseRvations and ExperimEnts) project, funded through the generosity of Eric and Wendy Schmidt by
recommendation of the Schmidt Futures program.




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





**Table 1 Regression models for each trait.** The multiple linear regression (MLR) model was fitted at community level, using only climate predictors. The Bayesian phylogenetic linear mixed model (BPLMM) was fitted at species level. The marginal $r^2$ includes climate effects only; the conditional $r^2$ also includes species identity and phylogeny as random effects. The phylogenetic comparative method (PCM), also fitted at species level, quantifies the variation attributed to climate alone, after factoring out effects of phylogenetic relatedness. The standard deviations of $r^2$ in BPLMM come from three different scenarios of phylogeny.

| Trait | BPLMM | | | | PCM | MLR |
|---|---|---|---|---|---|---|
| | marginal $r^2$ | conditional $r^2$ | species $r^2$ | phylogeny $r^2$ | $r^2$ | $r^2$ |
| $C_{mass}$ | 0.08 | 0.54±28 | 0.29±0.27 | 0.17±0.05 | 0.02 | 0.10 |
| $N_{mass}$ | 0.03 | 0.54±0.34 | 0.35±0.31 | 0.16±0.06 | 0.02 | 0.13 |
| C:N | 0.18 | 0.57±0.17 | 0.21±0.17 | 0.18±0.02 | 0.10 | 0.45 |





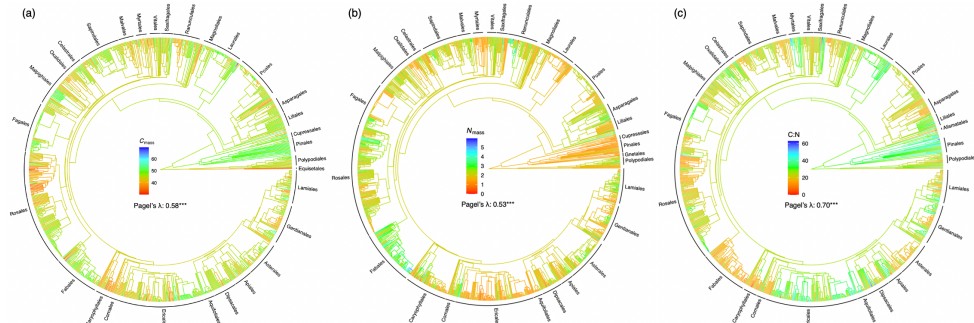

**Figure 1 Phylogenetic tree and signal for each leaf stoichiometric trait.** The colours of phylogenetic trees indicate trait value. Orders with more than five species in the data set are labelled. ***, $p < 0.001$.



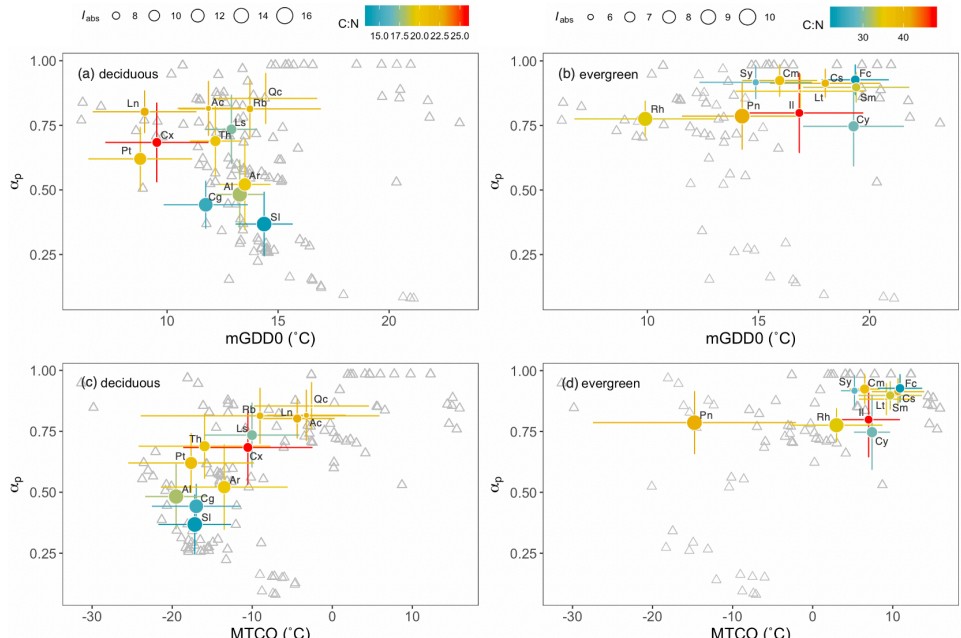

**Figure 2 Optima and tolerances of major genera in climate space.** $\alpha_p$ is a moisture index, mGDD0 is the mean temperature during the thermal growing season, MTCO is the mean temperature of the coldest month, and $I_{abs}$ is the leaf area index-weighted photosynthetic photon flux density. Colours of circles represent the values of leaf C:N ratio and sizes of circles represent $I_{abs}$. The grey triangles are sampling sites. Also shown are abbreviated names of genera (see Supplementary data Table S1).





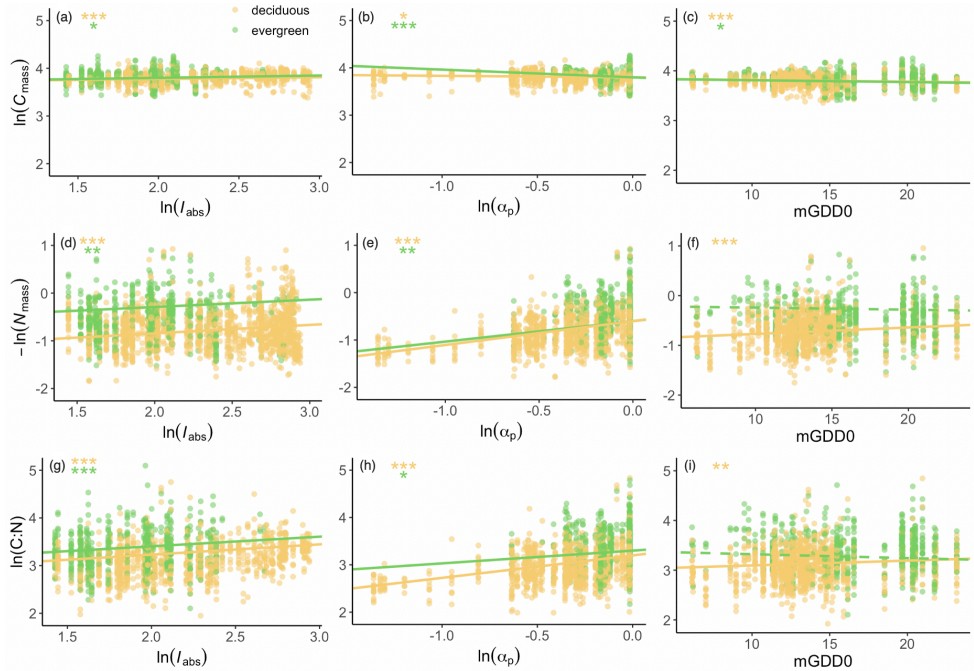

**Figure 3 Empirical partial relationships between leaf traits and climate.** $I_{abs}$ is the leaf area index-weighted photosynthetic photon flux density, $\alpha_p$ is a moisture index, and mGDD0 is the mean temperature during the thermal growing season. Yellow dots: deciduous species, green dots: evergreen species. The $N_{mass}$ scale is inverted, so that the slopes of the regression lines in panels (a)-(c) and (d)-(f) should add up to the slopes in panels (g)-(i).





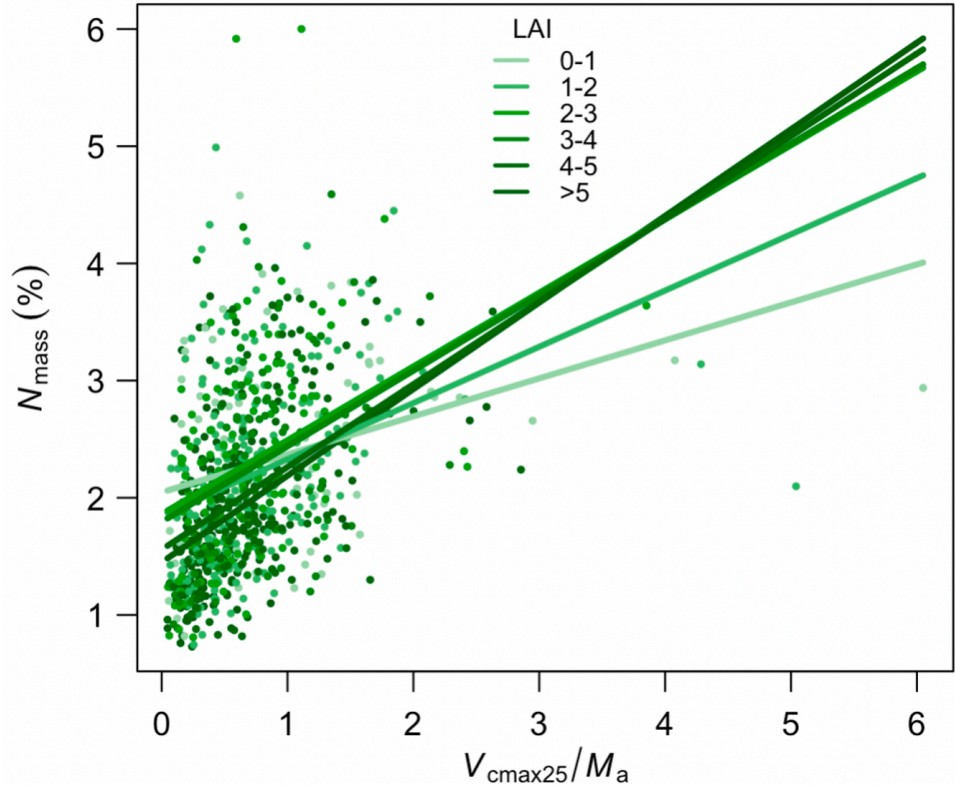

**Figure 4 The relationship between $N_{mass}$ and $V_{cmax25}/M_a$ along the leaf area index (LAI) gradient.** Colour saturation represents different levels of LAI. Lines are separate regressions for sites within each LAI bin.




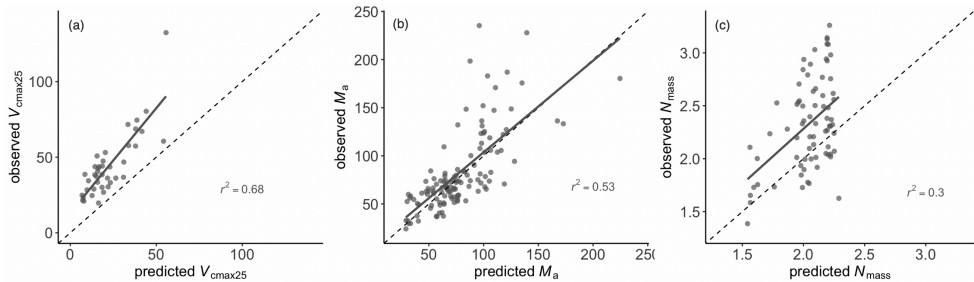

**Figure 5 Optimality-based predictions versus observations of leaf traits at site level.** Grey lines are ordinary least-squares regressions. The black dashed line is the 1:1 line.





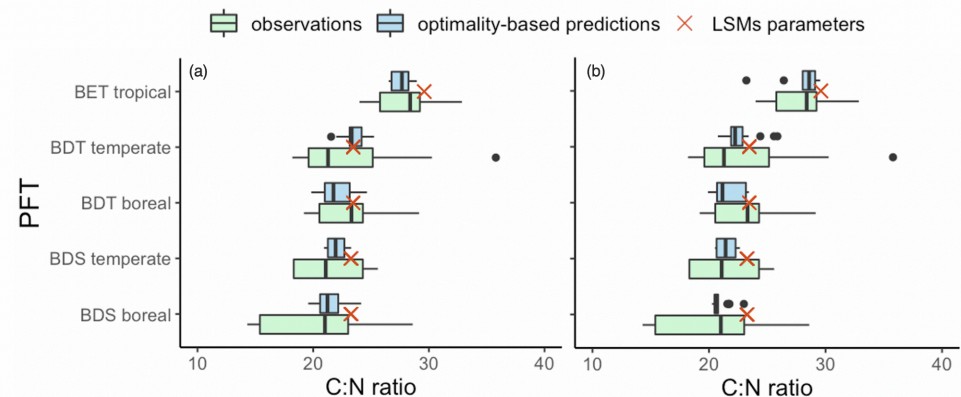


**Figure 6 Comparison of observed and predicted C:N ratios with target values in LSMs.** The blue boxes represent predicted C:N ratios using observed $C_{mass}$ (a), and using constant mean $C_{mass}$ (b). The green boxes represent observed C:N ratios. Red crosses show target values in LSMs.