# Peer review of "Leaf carbon and nitrogen stoichiometric variation along environmental gradients"

_Biogeosciences, 2023_

## Author Comment (AC1)

Summary:

In this manuscript, Xu et al demonstrate improving leaf C:N ratio representation in ESM by showing how environmental selection drives community leaf stoichiometry and individual plasticity plays a relatively small role.

The manuscript is very interesting and presents the problem and the authors approach well, but I have a question about the robustness of the analysis for Eco-Evolutionary Optimality as presented in the graphs. It looks to me like the main conclusions are affected by a low number of points with very high leverage. Can the results be presented to account for these outliers by log transforming the data or removing these points? A large part of the paper depends on accepting these analyses as robust. Correcting these may change some of the discussion.

**Authors:** We thank the reviewer for constructive and insightful comments. Our detailed answers are listed below.

**Reviewer:** Figure 4. These relationships are look like they are affected by a minority of points with a high VcMax25/Ma ratio.

**Authors:** We agree that the relationships shown in (the new) Figure 5 look as if they might potentially be biased by a few extreme values. However, we checked $V_{cmax25}$ and $M_a$ of the five species with exceptionally high $V_{cmax25}/M_a$ ratios. All but one were found to lie within expected ranges of $V_{cmax25}$ and $M_a$. Three out of five species with particularly high $V_{cmax25}$ (> 200 μmol m$^{-2}$ s$^{-1}$) were from high elevations (around 4000 m) where such values are both usual, and predictable (Wang et al., 2017 *New Phytologist*) due to the combination of high light intensity with low air pressure and temperature. At the 4081 m site (see Figure R1 below) half of all species had high $V_{cmax25}$ compared to other sites, but these still lay within the global range of $V_{cmax25}$ (Yan et al., 2023 *Global Ecology and Biogeography*). One species with particularly low $M_a$ (17.9 g m$^{-2}$) at the 1785 m site was nonwoody. Fig. R2 below shows $M_a$ varying from 11 to 147 g m$^{-2}$ at this site. Species with a similar range of $M_a$ values occurred at many other sites. Just one species at the 2258 m site showed exceptionally high $V_{cmax25}$ at moderate elevation. After we removed this one species, the result of the regression shifted only slightly, and the trend of increasing slope with LAI remained.

[Figure]

Figure R1 The distribution of $V_{cmax25}$ at sites with high $V_{cmax25}/M_a$ values and adjacent sites along elevation. The dots were individuals sampled at each site. Only sites along adjacent elevation gradients were shown as total number of sites was too many to make figure hard to read.

[Figure]

Figure R2 The distribution of $M_a$ at sites with high $V_{cmax25}/M_a$ values and adjacent sites along elevation. The dots were individuals sampled at each site. Only sites along adjacent elevation gradients were shown as total number of sites was too many to make figure hard to read.

**Reviewer:** Likewise in figure 5c, are the optimality predictions of Nmass skewed by the relatively low proportion of low Nmass species? It looks like the relationship would be very different without these points. Most of the species are between 2 and 2.25 with visually quite a different relationship.

**Authors:** $N_{mass}$ was indeed underpredicted and constrained within a narrow range, except for some sites (on the left side of the new Figure 6c) that were tropical seasonal forests, with

high LAI compared to other sites. When LAI was high, the intercept of the relationship between $N_{mass}$ and $V_{cmax25}/M_a$ was low (despite the steep slope) leading to low predicted $N_{mass}$. We have provided further discussion about our method to predict $N_{mass}$ and its potential improvement (Lines 336-340):

"However, our predicted $N_{mass}$ was constrained within a narrow range, despite the well-captured variations in $M_a$ and $V_{cmax25}$. The predicted $N_{mass}$ in tropical forests with high LAI were systematically underestimated due to the low intercept (Supplementary data Table S1). We recognize that our method to predict $N_{mass}$ may overlook additional functions of N in leaves, such as chemical defences, perhaps causing greater variation than predicted. This requires further investigation."

**Reviewer:** A simple explanation of what exactly Pagels λ is – what is phylogentetic signal – would be useful to readers with a more biogeochemical background as one would expect from this journal

**Authors:** We have added some explanation about Pagels λ (Lines 156-159):

"Phylogenetic signal was calculated for each trait, using Pagel's λ, which measures the extent to which related species tend to have similar trait values. Pagel's λ varies from 0 to 1, indicating low to high phylogenetic signal. It was calculated using the `phytools` package (Münkemüller et al., 2012; Revell, 2012). The significant values obtained indicate that values of these traits tend to be conserved within lineages."

**Reviewer:** Why specifically is the China plant trait database used? What advantage is this giving over other trait databases? Given that the rationale is improving models, would a global database be more suitable? Also, if I understand it correcvtly, the physical sampling methods described L76 – 89 are the direct collation of this database? This could be clearer.

**Authors:** The unique advantage of the China Plant Trait Database is that it provides data from the same populations, sampled at the same time, for $M_a$, $V_{cmax25}$, $N_{mass}$ and χ. This was indispensable for our analysis.

We have revised our description of the sampling method for greater clarity (Lines 79-81).

"In CPTDv2, a stratified sampling strategy was consistently used at each site to ensure that the dominant species in each canopy layer were sampled (detailed in Wang et al. (2018)) and avoid bias of different sampling strategies."

**Reviewer:** L61 – this sentence is quite unclear, not sure if the reference temp of 25 C refers to Vcmax25 or both this and Ma

**Authors:** The reference temperature of 25˚C only refers to $V_{cmax}$, not $M_a$. We have revised the sentence to make this clear:

"We assumed that the metabolic and structural components of leaf N are proportional to carboxylation capacity ($V_{cmax25}$, at a reference temperature of 25˚C) and $M_a$, respectively."

**Reviewer:** L102 – individuals of the same species? Or different species for community averages? If so, how were these determined?

**Authors:** For each species at a site, leaf C content, N content and $\delta^{13}$C were measured using three or more individuals of the same species. The community means of traits were averages of all species at a site. We have now revised this description as follows:

"For each species at a site, leaf C content, N content and $\delta^{13}$C were measured using pooled samples of leaves from at least three individuals of the same species."

**Reviewer:** L205 – are these fixed values the same across all LSMs? This is unclear to me from the text and from Figure 6.

**Authors:** The fixed values are almost the same across several LSMs, including CLM4, ED2.1, JSBACH and ORCHIDEE. We have added clarification in the text and figure.

"The target (PFT-specific) values used in several LSMs such as CLM4, ORCHIDEE and YIBs (Fig. 7) are based on datasets nearly 20 years old and fail to represent continuous trait variations that can now be inferred from much larger data sets."

**Reviewer:** Figure 1 – this figure is really hard to read, it needs to be larger or simpler.

**Authors:** We have made the text in the figure larger.

**Reviewer:** Figure 2 – with 11 genera, this could be listed in the caption and reduce reliance on the SI

**Authors:** We have now put all the information in the caption.

**Reviewer:** Figure 3 – caption should indicate what the *** mean

**Authors:** We have added its meaning in the caption.

---

## Author Comment (AC2)

General comments

This manuscript with reference ID bg-2023-87 presents an optimality-based approach investigating the drivers of leaf trait variation along elevational gradients across China. To this end, the authors make use of data available from the China Plant trait database (version 2) to parameterize and test an eco-evolutionary optimality model for leaf nitrogen per unit mass (Nmass). Results obtained with a Bayesian phylogenetic linear mixed model suggest that variation in leaf stoichiometric traits are mainly controlled by species identity and phylogeny, thus indicating that accounting for community level responses and shifts in species turnover may allow for a more dynamic representation of ecosystem processes in Earth System models. Albeit the fact that this conclusion is not novel, the analysis appears to be sound and the manuscript is concise and very well written. Hence, I conclude that the article should be of great interest to the academic readership of the journal, and subject to minor amendments and modifications (see recommendations in the specific comments provided below), could be considered for publication.

**Authors:** We appreciate this comprehensive and encouraging evaluation.

Specific comments

The study by Xu and colleagues presents an interesting analysis investigating the drivers of leaf trait variation across environmental gradients. While the manuscript is generally well written and the findings are presented in a concise and informative way, I would suggest adding some further clarifications with regard to (i) statistical analysis, (ii) intra-specific trait variation, and (iii) parameters obtained from remote sensing.

**Reviewer:** First, there appears to be a potential spatial bias in the analysis (L71-74) investigating trait variation across large spatial scales, such as the large-scale environmental gradients across China. For instance, multiple regression on distance matrices (MRM) could be applied to quantify the relative amount of trait variation in response to space and environmental factors in ecological data (Lichstein, 2006) and to relate phylogenetic or functional beta diversity to spatial and environmental distance (Swenson, 2014).

**Authors:** We have implemented MRM to account for the spatial effect on trait variation, and added this it into the results (Line 151-154; Supplementary data Table S2). We found that leaf stoichiometric traits were not significantly related to spatial distance, but were strongly explained by climatic factors. We have summarized this finding as follows (Lines 204-206):

"MRM analysis also showed that trait variations were strongly explained by climatic factors, but not significantly related to geographic distance – indicating that the purely spatial effect on trait values is weak (Supplementary data Table S2)."

**Reviewer:** Second, the lack of phenotypic plasticity in leaf stoichiometry (L237-239) and the associated conclusion that leaf stoichiometric traits might be mainly controlled by species identity and phylogeny without proper consideration of intraspecific trait variation (ITV) could be misleading as the mechanisms driving trait variation across environmental gradients have been reported to shift across large spatial gradients (Ackerly & Cornwell, 2007). Whereas, across larger spatial scales abiotic factors, such as temperature and precipitation, represent

key determinants of ecosystem processes, at smaller spatial scales other biotic factors, such as competition among coexisting tree species, strongly affect ecosystem structure and functioning via the composition of the local species pool (Hofhansl, 2021). Hence, biotic factors can have equally strong impacts on trait expression as the dominant abiotic driver (Albert, 2010; Jung 2010, Violle 2012). As a result, an increasing number of studies documented the importance of ITV and thus it would be great to see a discussion on the potential of including ITV in optimization-based models, such as the one applied in this study.

**Authors:** We fully agree about the importance of intraspecific variation along trait gradients. To address this comment, we have implemented trait-gradient analysis in order to quantify intraspecific trait variation (Lines 174-180; 196-200), and added discussion of this result (Lines 340-346):

"Trait-gradient analysis showed that in addition to species turnover, intraspecific trait variation plays a role in determining trait shifts at a regional scale (Fig. 3). The intraspecific slopes for $C_{mass}$, $N_{mass}$ and their ratio were calculated for 19, 19 and 42 species respectively. Only 9, 8 and 16 of these species showed significant slopes. The intraspecific slopes for $N_{mass}$ and C:N ratio ranged from 0.7 to 2.1 and 0.6 to 1.9, respectively. The slopes for $C_{mass}$ ranged from 0.8 to 1.4 except for one species (*Asparagus dauricus*) that had a negative slope."

"Systematic variation in community-mean leaf stoichiometric traits along climate gradients can be achieved through progressive species replacement at a macroclimatic scale, and intraspecific trait variability at a regional scale (Liu et al., 2019; Yang et al., 2016). Some species in this study apparently adjusted their leaf stoichiometry along major environmental gradients, possibly via genetic adaptation over multigenerational timescales. Due to the lack of intraspecific data within communities, we could not assess the degree of variation among conspecific plants in the same environment. Intraspecific variation within communities may however increase functional diversity and promote species coexistence (Westerband et al., 2021) and potentially provide a buffer against climatic variation and change (Ahrens et al., 2021). Further studies are needed to better understand intraspecific trait variation (Moran et al., 2016) in order to assign appropriate timescales for the dynamic responses of traits to environmental changes in Earth system models."

**Reviewer:** Third, the lack of a significant relationship of leaf stoichiometry with LAI and soil fertility (L252-253) both obtained from remote sensing estimates, and the controversial finding that nitrogen allocation to metabolic and structural components was related to leaf area index (L324-325), made me wonder if it would actually require data obtained from in-situ measurements (that match the spatial and temporal extend of the trait data) in order to identify these effects.

**Authors:** No LAI or soil information was collected during sampling, so it was necessary to rely on remotely sensed LAI and global soil products – a strategy which, we agree, carries significant uncertainty. We have pointed this out in our revised discussion (Lines 278-280; 336-340):

"Uncertainty in our soil fertility data may was inevitably introduced due to our reliance on a gridded soil map (Shangguan et al., 2013). More studies including *in situ* soil measurements

are needed to more comprehensively investigate the effect of soil properties on plant stoichiometry."

"...our predicted $N_{mass}$ was constrained within a narrow range, despite the well-captured variations in $M_a$ and $V_{cmax25}$. The predicted $N_{mass}$ in tropical forests with high LAI were systematically underestimated due to the low intercept (Supplementary data Table S1). We recognize that our method to estimate $N_{mass}$ may overlook additional functions of N in leaves, such as chemical defences, perhaps causing greater variation than predicted. This requires further investigation."

**Reviewer:** Overall, I would appreciate a more thorough discussion on some of the topics indicated above and would therefore recommend revising the manuscript based on the findings presented in the scientific literature (see additional references to be considered below) and how these results could be used to improve the dynamic representation of plant tissue stoichiometry in Earth System models.

**Authors:** We have added discussion about the role of intraspecific trait variation in particular. Please see above.

**Reviewer:** L316: correct typo: "our EEO-based approach thus suggests …"

**Authors:** We have corrected this in the text.

**Reviewer:** L613-615: Please add a description for the (i) labels "Cmass", "Nmass", "C:N ratio"; (ii) colour code (red-blue gradient); and (iii) test statistics used in respective panels of Figure 1 A/B/C.

**Authors:** We have added this information in the caption.